Remarkable genomic diversity among Escherichia isolates recovered from healthy chickens

Thomson Nicholas M. 1
Gilroy Rachel 1
http://orcid.org/0000-0002-2937-3420 Getino Maria 2 3
http://orcid.org/0000-0001-6620-9403 Foster-Nyarko Ebenezer 1 4
van Vliet Arnoud H.M. 3
La Ragione Roberto M. 3 5
http://orcid.org/0000-0003-1807-3657 Pallen Mark J. 1 3 6 mark.pallen@quadram.ac.uk
1 Quadram Institute Bioscience , Norwich, Norfolk , United Kingdom
2 NIHR Health Protection Research Unit in Healthcare Associated Infections and Antimicrobial Resistance, Department of Infectious Disease, Imperial College London , London , United Kingdom
3 Department of Pathology and Infectious Diseases, School of Veterinary Medicine, University of Surrey , Guildford, Surrey , United Kingdom
4 Department of Infection Biology, London School of Hygiene & Tropical Medicine, University of London , London , United Kingdom
5 Department of Microbial Sciences, School of Biosciences and Medicine, University of Surrey , Guildford, Surrey , United Kingdom
6 School of Biological Sciences, University of East Anglia , Norwich, Norfolk , United Kingdom
Thomas Jonathan
Electronic publication date: 2022 Mar 1
Publication date: 2022
Volume: 10
Electronic Location ID: e12935
Received 2021 Nov 12; Accepted 2022 Jan 23
Copyright: © 2022 Thomson et al.
Copyright year: 2022
Copyright holder: Thomson et al.
License: This is an open access article distributed under the terms of the Creative Commons Attribution License, which permits unrestricted use, distribution, reproduction and adaptation in any medium and for any purpose provided that it is properly attributed. For attribution, the original author(s), title, publication source (PeerJ) and either DOI or URL of the article must be cited.
License URL: https://creativecommons.org/licenses/by/4.0/

Keywords: Escherichia, Genomic diversity, Chickens, Phylogenomics, Commensal, Cryptic clades

Funding: BBSRC Institute Strategic Programme Microbes in the Food Chain BB/R012504/1 and constituent project BBS/E/F/000PR10351 European Joint Programme One Health EJP EU Horizon 2020 Research and Innovation Programme 773830 Nicholas M. Thomson, Rachel Gilroy and Mark J. Pallen were supported by the BBSRC Institute Strategic Programme Microbes in the Food Chain BB/R012504/1 and its constituent project BBS/E/F/000PR10351. Maria Getino, Arnoud H.M. van Vliet and Roberto M. La Ragione were supported by the European Joint Programme One Health EJP. This project received funding from the European Union’s Horizon 2020 research and innovation programme under Grant Agreement No. 773830. There was no additional external funding received for this study. The funders had no role in study design, data collection and analysis, decision to publish, or preparation of the manuscript.

==============================
The genus Escherichia has been extensively studied and it is known to encompass a range of commensal and pathogenic bacteria that primarily inhabit the gastrointestinal tracts of warm-blooded vertebrates. However, the presence of E. coli as a model organism and potential pathogen has diverted attention away from commensal strains and other species in the genus. To investigate the diversity of Escherichia in healthy chickens, we collected fecal samples from antibiotic-free Lohmann Brown layer hens and determined the genome sequences of 100 isolates, 81 of which were indistinguishable at the HC0 level of the Hierarchical Clustering of Core Genome Multi-Locus Sequence Typing scheme. Despite initial selection on CHROMagar Orientation medium, which is considered selective for E. coli, in silico phylotyping and core genome single nucleotide polymorphism analysis revealed the presence of at least one representative of all major clades of Escherichia, except for E. albertii, Shigella, and E. coli phylogroup B2 and cryptic clade I. The most frequent phylogenomic groups were E. coli phylogroups A and B1 and E. ruysiae (clades III and IV). We compiled a collection of reference strains isolated from avian sources (predominantly chicken), representing every Escherichia phylogroup and species, and used it to confirm the phylogeny and diversity of our isolates. Overall, the isolates carried low numbers of the virulence and antibiotic resistance genes typically seen in avian pathogenic E. coli. Notably, the clades not recovered are ones that have been most strongly associated with virulence by other studies.

Introduction

Members of the genus Escherichia are common inhabitants of the gastrointestinal tracts of warm-blooded animals (Tenaillon et al., 2010; Jang et al., 2017). Although many isolates are believed to be beneficial or harmless components of a healthy microbiome, the genus has received a large share of research attention as some members—notably certain strains of Escherichia coli—cause disease and/or carry transferable antibiotic resistance genes (Kaper, Nataro & Mobley, 2004; Cummins et al., 2019).

Much effort has been expended to develop techniques for identifying and categorizing members of Escherichia and understanding the population structure of the genus. Classical phenotypic techniques identified species by morphology and biochemical tests and divided strains into pathotypes, based on site of infection, e.g., uropathogenic E. coli (UPEC) causing urinary tract infections; and serotypes, based on antibody recognition of variation in lipopolysaccharides, flagella and fimbriae (Fratamico et al., 2016). These methods have gradually given way to genome-based approaches classifying isolates with multi-locus enzyme electrophoresis studies (Selander et al., 1987) and multi-locus sequence typing (Maiden et al., 1998). Whole genome sequencing technologies have recently facilitated the comparison of strains in ever greater detail (Uelze et al., 2020), leading to new classification schemes based on average nucleotide identity (Konstantinidis & Tiedje, 2005) and whole-genome nucleotide polymorphism (Schürch et al., 2018).

The avian pathogenic E. coli (APEC) pathotype causes colibacillosis in chickens, turkeys and other avian species, and is responsible for significant morbidity and mortality in the worldwide poultry industry (Dziva & Stevens, 2008). Certain serotypes and virulence factors are frequently associated with APEC but it remains challenging to distinguish categorically between commensal and virulent strains (Mehat, van Vliet & La Ragione, 2021).

The large number of high-quality Escherichia genomes from diverse geographical and biological sources has forced a re-thinking of classification. Thus, although Shimwellia blattae was initially placed within Escherichia, it is now assigned to a separate genus (Priest & Barker, 2010). By contrast, although named as if a genus, Shigella is now thought merely to represent a series of pathovars of E. coli (Yang et al., 2005; Devanga Ragupathi et al., 2018). Nevertheless, the assignment of strains of E. coli to seven phylogroups (A, B1, B2, C, D, E and F) has proven a robust finding that holds for phylogenetic trees built from core genome alignments of large numbers of isolates. Phylogroups G and H have recently been proposed (Lu et al., 2016; Clermont et al., 2019), while five of what were initially called ‘cryptic clades’ (Walk et al., 2009) have been assigned to species: Clade I to E. coli (Clermont et al., 2013), Clade II to E. whittamii (Gilroy et al., 2021), Clades III and IV together to E. ruysiae (van der Putten et al., 2021), and Clade V to E. marmotae (Liu et al., 2015).

The accumulation of genomic data for Escherichia has been strongly biased towards clinical isolates from humans and economically important animals (Touchon et al., 2020). This hampers efforts to achieve a complete understanding of Escherichia diversity, ecology, and population genetics. Here, we report phylogenomic analyses on 100 Escherichia isolates cultured from fecal samples from a small flock of healthy adult layer chickens, 81 of which we determined to be unique (Fig. 1)

Figure 1 Summary of the major stages of sample collection, processing and data analysis and the primary programs and databases used.

.

Materials and Methods

Sample collection and storage

The project was approved by the University of Surrey’s NASPA Ethical Review Assessment Committee with project number NERA-2018-011. Fresh fecal samples from healthy adult Lohmann Brown layer hens on a farm in the south-east of England were collected to isolate presumptive E. coli. The birds (flock size = 24) were purchased from a single flock in April 2017 at 20 weeks old and kept in a large outdoor run with a substrate of stone chippings and small turf enrichment beds during the day and in a coop overnight. They were fed Farmgate Layer pellets and Mash (ForFarmers Ltd, Bury St Edmunds, UK), according to the manufacturer’s instructions, and no antibiotics were used. Sampling was carried out between July and November 2018. Once a month, five freshly voided fecal samples were collected from different hens. 1:10 serial dilutions of each sample were plated (100 μL) onto MacConkey agar to obtain single colonies. Several colonies with different morphologies were selected and streaked to fresh MacConkey agar plates to isolate single colonies. Identification as presumptive E. coli was confirmed by a purple colony phenotype on CHROMagar Orientation Medium (CHROMagar, Paris, France) and a positive indole test. Multiple E. coli-positive isolates from each sample were tested by GTG5 PCR (Mohapatra, Broersma & Mazumder, 2007) to minimize the collection of identical genotypes. Isolates were stored using the Microbank Bacterial and Fungal Preservation system (Pro-Lab Diagnostics, Richmond Hill, Canada) at −80 °C.

Antimicrobial resistance screening

Isolates were screened for resistance against a panel of eight antibiotics (gentamicin, ampicillin, trimethoprim, chloramphenicol, nitrofurantoin, cefpodoxime, meropenem and ciprofloxacin) by the disk diffusion method, according to EUCAST guidelines (https://www.eucast.org/ast_of_bacteria). They were additionally subjected to a minimum inhibitory concentration assay for colistin by broth dilution, according to EUCAST guidelines.

DNA extraction and genome sequencing

A total of 100 isolates were submitted for whole genome sequencing, comprised of multiple isolates per fecal sample from the first 3 months and one isolate per sample for the remaining 2 months (Table S1). Genomic DNA was extracted from overnight 1 mL cultures in Lysogeny Broth (LB; Miller’s formulation), maintained at 37 °C statically in sterile deep-well plates prepared from single colonies taken from overnight Lysogeny Agar plates, according to a previously described 96-well plate lysate method (Foster-Nyarko et al., 2020). Briefly, cultures were pelleted at 3,500 × g then resuspended and pre-treated with lysozyme, proteinase K and RNase A before lysing with 10% (w/v) sodium dodecyl sulphate in Tris-EDTA (pH 8.0). The DNA was retained on AMPure XP solid-phase reversible immobilization beads (Beckman Coulter, High Wycombe, UK), eluted in Tris/HCl, pH 8.0 and quantified using the Qubit high-sensitivity double-stranded DNA assay (Invitrogen, MA, USA).

For sequencing, samples were normalized to 0.5 ng.µL−1 with 10 mM Tris-HCl prior to library preparation, which was carried out with a modified Nextera XT DNA protocol (Foster-Nyarko et al., 2020). The pooled libraries were run at a final concentration of 1.8 pM using the Illumina NextSeq 500 platform (Illumina, CA, USA) following the manufacturer’s recommended denaturation and loading procedures, which included a 1% PhiX spike, with 150 bp paired-end reads.

Genome assembly

Raw sequence data were demultiplexed and converted to fastq format by bcl2fastq v.2.20 (Illumina, CA, USA). Reads from multiple lanes were concatenated into single forward and reverse read files then uploaded to Enterobase (www.enterobase.warwick.ac.uk) for automated quality control and assembly via the QAssembly pipeline (Zhou et al., 2020). All assemblies exceeded the minimum quality requirements for Enterobase (genome length between 3.7–6.4 Mbp; N50 > 20 kb; number of contigs ≤ 800; proportion of Ns < 3%; >70% of contigs assigned correctly by Kraken 2 (Wood, Lu & Langmead, 2019)). The final assemblies were downloaded for further analyses, which were all run on the Cloud Infrastructure for Microbial Bioinformatics (Connor et al., 2016).

Phylogenomic analysis

Initial phylogenomic analysis was carried out in Enterobase for in silico serotyping, fimH allele typing (Roer et al., 2017), multi-locus sequence typing according to the Warwick seven-gene scheme and phylotyping (Beghain et al., 2018), and to construct neighbor-joining trees with the NINJA algorithm in GrapeTree (Zhou et al., 2018), based on the Hierarchical Clustering of Core Genome MLST (HierCC) scheme (Zhou et al., 2020). HierCC was also used to de-replicate the strain collection by selecting a single representative isolate from every identical cluster of genomes at the HC0 level. Representative isolates were determined according to the quality score-based system employed as part of dRep v2.5.4, reliant on genome completion, contamination and N50 metrics (Olm et al., 2017). The assemblies for the 33 strains involved in clustering were used to construct a core genome alignment using Snippy v.4.3.2 (https://github.com/tseemann/snippy). A matrix of pairwise single nucleotide polymorphisms (SNPs) was then compiled with snp-dists v.0.7.0 (https://github.com/tseemann/snp-dists). E. coli MG1655 (NC_000913.3) was used as the reference.

To confirm the identities and phylogenetic relationships of the final isolate collection, a second core genome alignment was reconstructed, containing all strains. In addition, we collated a panel of reference genomes (Table S2) for every phylogroup of E. coli and all other Escherichia species, including Shigella flexneri, all of which had been isolated from avian hosts (primarily chickens). These reference genomes were downloaded from the NCBI assembly archive and included in the core genome alignment, which was then used to reconstruct a core SNP phylogenetic tree with Salmonella bongori and Salmonella enterica as outgroups. IQ-TREE v.2.0.3 (Nguyen et al., 2015) was used with 1,000 bootstrap replications for maximum-likelihood inference of phylogenetic relationships with the best fitting model (TVM+F+ASC+R4) automatically selected by ModelFinder (Kalyaanamoorthy et al., 2017). The resulting tree was visualized and combined with other data using iTOL v.5 (Letunic & Bork, 2021).

Analysis of gene and plasmid content

To process shared gene content across our selected genome catalogue, we used the pangenomics pipeline (Delmont & Eren, 2018) as implemented in anvi’o v.7.0 (Eren et al., 2015) with open reading frames predicted using prodigal v.2.6 (Hyatt et al., 2010) and annotation using the NCBI’s Clusters of Orthologous Groups database (Tatusov et al., 2003). Using NCBI BLAST, the similarity between gene pairs was quantified and subsequently the Markov Cluster algorithm determined clusters of homologous genes, with a minbit heuristics threshold of 0.5 to eliminate weak matches and an MCL inflation of 10 for closely related genomes. FastANI (Jain et al., 2018) was applied for the calculation of average nucleotide identity between genomes. The program anvi-display-pan provided an interactive visualization of the pangenome, with imported average nucleotide identity values being depicted as part of this visualization.

ABRicate v.1.0.1 (https://github.com/tseemann/abricate) was used to search assemblies for genes related to antibiotic resistance and virulence and for plasmid replicons by comparison to the NCBI AMRFinderPlus (https://www.ncbi.nlm.nih.gov/bioproject/PRJNA313047), ecoli_VF (https://github.com/phac-nml/ecoli_vf) and PlasmidFinder (Carattoli et al., 2014) databases, respectively. A custom database was prepared for detection of 24 APEC-related virulence genes based on a literature search (File S1). In each case, identification was defined by minimum coverage of 90% and minimum identity of 80% of the respective nucleotide sequences. To detect plasmid-encoded virulence-associated genes we first identified contigs derived from plasmids using platon v.1.6 in accuracy mode (Schwengers et al., 2020), then searched those contigs against our custom database using ABRicate as described above.

Results

High phylogenomic diversity of isolates recovered

One hundred isolates provisionally identified as E. coli were collected from healthy layer hens over a 5-month period. An indication of the diversity of Escherichia recovered was provided by the isolation of 1–7 isolates from up to six phylogroups per fecal sample (Table 1). When cultured on MacConkey agar, 15 isolates were non-lactose-fermenters but positive for other E. coli-specific attributes (Nicoletti et al., 1988).

Table 1 Collection dates, age of birds and number of isolates recovered for each fecal sample.

Faecal sample	Date collected	Age of birds (weeks)	GTG5 unique isolates	HC0 unique isolates	Phylogroups	
1	09/07/2018	84	5	5	A, B1	
2	09/07/2018	84	6	5	A, B1, (II)	
3	09/07/2018	84	6	6	A, B1, C	
4	09/07/2018	84	6	3	A, Ef	
5	09/07/2018	84	6	4	A, D, (V)	
6	13/08/2018	89	7	7	A, B1, F, Ef, (III), (V)	
7	13/08/2018	89	5	4	A, B1	
8	13/08/2018	89	6	5	A, Ef, (III), (V)	
9	13/08/2018	89	6	6	A, B1, (V)	
10	13/08/2018	89	5	3	B1, (III), (IV)	
11	10/09/2018	93	6	6	A, B1, E, Ef, (III)	
12	10/09/2018	93	7	5	A, B1, E, Ef	
13	10/09/2018	93	7	5	A, B1, (III)	
14	10/09/2018	93	7	4	A, (IV), (V)	
15	10/09/2018	93	6	4	A, (III), (V)	
16	08/10/2018	97	1	1	B1	
17	08/10/2018	97	1	1	(IV)	
18	08/10/2018	97	1	1	(III)	
19	08/10/2018	97	1	1	A	
20	08/10/2018	97	1	1	D	
21	12/11/2018	102	1	1	(V)	
22	12/11/2018	102	1	1	D	
23	12/11/2018	102	1	1	B1	
24	12/11/2018	102	1	1	A	
			100	81		
Note:

Following culturing on MacConkey agar and CHROMagar Orientation medium, 100 presumptive E. coli were isolated. Multiple colonies per fecal sample were screened by GTG5 PCR to reduce the collection of identical strains. Whole-genome sequence analysis revealed the large diversity of Escherichia present despite apparently selecting for E. coli, although only 81 of the initial strains were distinguishable at the HC0 level using HierCC clustering (see Materials and Methods). Phylogroups of E. coli are referred to by individual letters; cryptic clades are referred to by Roman numerals in parentheses; Ef, E. fergusonii.

Following short read whole genome shotgun sequencing and assembly, the HierCC feature within Enterobase identified 33 out of the 100 isolates as belonging to one of 14 clusters at the HC0 level, meaning that the members of each cluster are indistinguishable from each other at every core genome locus interrogated (Table 2). The 14 HC0 clusters all contained isolates taken from different fecal samples, spanning the duration of the study, except cluster HC0:148574, which contained three isolates that all came from a single sample taken in September 2018. A core genome alignment of the 33 strains involved in clustering was used to construct a matrix of pairwise SNP distances, confirming the close relationships between clustered strains, with 2–257 SNPs detected (Table 2, Table 3). Since we were primarily interested in assessing species diversity within our samples, we therefore removed 19 isolates from the analysis by keeping only the best quality assembly from each HC0 cluster as a representative (Table S3).

Table 2 Clustering of isolates by HierCC level HC0 and selection of representative isolate for each cluster.

HC0 cluster	MLST	Isolates	Core SNP distance	Representative isolate	
148525	5643	021, 043	124	021	
148529	155	007, 037	21	037	
148537	48	024, 032	11	032	
148541	1112	026, 067	6	067	
148542	1638	006, 055	11	006	
148547	5573	025, 041, 057, 073, 087	See Table 3	025	
148548	1276	034, 089	18	034	
148555	48	018, 042	20	042	
148556	7059	045, 085	55	045	
148564	1844	019, 050, 078	See Table 3	050	
148571	6540	063, 080	257	063	
148574	2456	068, 069, 071	See Table 3	068	
148603	48	004, 081	2	004	
148607	11513	020, 058	143	058	

Table 3 SNP distance matrices for HC0 clusters of >2 isolates.

SNP distance matrix for Cluster HC0_148547	
Strains	041	057	073	087	
025	9	9	9	4	
041	0	5	11	11	
057		0	13	8	
073			0	10	
SNP distance matrix for Cluster HC0_148564	
	050	078			
019	16	15			
050	0	14			
SNP distance matrix for Cluster HC0_148574	
	069	071			
068	8	14			
069	0	10			
Note:

Table 2 provides SNP distances for 11 HC0 clusters comprising pairs of isolates. The remaining three clusters contained >2 isolates, requiring a matrix for full pairwise comparison.

In silico evaluation of Warwick seven-gene MLSTs for the remaining 81 isolates revealed the presence of 45 different sequence types, of which 30 were represented by a single isolate each. ST48 (nine isolates), ST10 (seven isolates) and ST2456 (five isolates) were the most numerous. HC0 isolate clusters correlated with MLSTs, supporting the close relationship between strains in each cluster (Table S1). Predicted phylogroups also suggested considerable genetic diversity among the isolates, with at least one representative from all seven E. coli phylogroups except B2, every ‘cryptic clade’ except clade I, and five E. fergusonii (Table 1, Fig. S1). In silico prediction of O:H serotype and fimH alleles also pointed towards considerable diversity, with 30 different O antigens, 22 different H antigens and 24 different fimH alleles detected (Table S1). Isolates outside of E. coli sensu stricto had increased incidence of ‘undetermined’ results for O antigen and fimH, suggesting that current databases do not cover the whole diversity of these features for the genus. The diversity of strains did not have a significant temporal component as isolates collected in different months were evenly distributed throughout a neighbor-joining phylogenetic tree based on the HierCC results (Fig. S1).

A core SNP maximum likelihood tree containing the 81 isolates and reference genomes for every clade of Escherichia was in good agreement with the neighbor-joining tree, and the predicted phylogroups clustered with their respective reference genomes (Fig. 2). This validates the reference genome collection, indicating its value for providing a scaffold phylogeny for future studies. Three isolates that were identified as ‘group E or clade I’ by in silico phylotyping were assigned to phylogroup D based on the core SNP alignment. Isolate Surrey070 was also reassigned, as it clustered more closely with phylogroup E than its predicted phylogroup of D.

Figure 2 Core SNP maximum likelihood tree of the final 81 Escherichia isolates, including reference sequences for every Escherichia species and all phylogroups of E. coli.

Names of our isolates and their reference strains are coloured according to their species/phylogroup. Names of reference strains for clades not found in our samples are coloured black. Reference strains are named with their species and strain name, with phylogroups given in parentheses. Salmonella bongori and Salmonella. enterica ser. Typhi are included as outgroups.

The pangenome of isolates reflects phylogenomic diversity

More than 500,000 genes were identified within the 81 study isolates and 33 reference genomes before organization into 14,421 gene clusters, formed by grouping homologous genes according to amino acid similarity, with a minbit heuristics threshold of 0.5 and an MCL inflation of 10. The core genome (genes present in all genomes) contained 2,449 gene clusters and the accessory genome (genes present in >1 but not all genomes) contained 8,331 gene clusters, while 3,641 singleton clusters (present in only one genome) were identified. All accessory and core genes were hierarchically clustered according to distribution pattern (Fig. 3). Functional hits were assigned to gene clusters according to NCBIs Clusters of Orthologous Groups database, with 2,766 unique groups annotated (Table S4). A total of 94% of core gene clusters were annotated compared to the considerably lower proportion annotated for both accessory (50%) and singleton (30%) gene clusters. Functional pathways assigned to annotated gene clusters were primarily associated with translation, ribosomal structure, and biogenesis (9.5% of annotated core gene clusters) with accessory gene clusters associated with prophages and transposons (9.8% of annotated accessory gene clusters). Both core and accessory splits had prominent proportions of gene clusters annotated with carbohydrate transport and metabolism functions (7.2% and 10.1% of annotated gene clusters, respectively).

Figure 3 Anvi’o representation of the Escherichia pangenome showing 81 Escherichia isolates recovered from chicken faeces and 33 publicly available reference Escherichia isolates from poultry species.

Each layer represents a single genome, with black colouring signifying the presence of a gene cluster. Gene clusters are organised according to their distribution across the genome, with co-occurring genes shown closer together. The heatmap at the top-right of the image represents average nucleotide identity across all included genomes, with darker red colours indicating a higher percentage of average nucleotide identity. Assigned phylogroup is shown by the colour bar on the right of the image. Singleton gene clusters (present only in one genome; n = 3,641) are highlighted in blue while core gene clusters (present in all genomes; n = 2,449) are shown in green.

Plasmid incompatibility groups are shared throughout Escherichia

A total of 25 different plasmid replicons were identified by comparison with the PlasmidFinder database (Table S5). Eight strains did not contain a plasmid replicon, while two strains (Surrey037 and Surre074) contained seven different replicons (Fig. 4). The overall rate of plasmid carriage was low, with a mode of 1 replicon and median of 2. There were 14 instances of an isolate containing multiple replicons of the same identity, although these were confined to the Col(MG828)_1, Col440I_1 and ColRNAI_1 groups. The most common replicons were ColRNAI_1 (46 replicons in 35 isolates) and p0111_1 (36 replicons in 36 isolates). Of the broad plasmid incompatibility groups, Col plasmids were most frequently found (101 replicons), partly because of their propensity for multiple replicons per isolate, with IncF (58 replicons) and p0111_1 (36 replicons) also highly represented (Fig. 5). There were no clear associations between plasmid incompatibility groups and phylogenomic groups, which is consistent with frequent transfer of genetic material within the genus (Shaw et al., 2021).

Figure 4 Summary of the plasmid replicon contents of 81 Escherichia isolates.

(A) Number of isolates containing 0–7 distinct plasmid replicons from the PlasmidFinder database. (B) Frequency of each plasmid replicon identified from the PlasmidFinder database. Some isolates contained >1 of the same replicon type (see main text).

Figure 5 Presence (solid squares) and absence (open squares) of predicted antibiotic resistance genes, virulence-associated genes and plasmid replicons in the final 81 isolates.

Isolates are arranged according to the phylogenetic tree shown in Fig. 2 with their names coloured according to phylogenetic group. Ew, E. whittamii; Em, E. marmotae; Er, E. ruysiae; Ef, E. fergusonii; A–F, E. coli phylogroups. Virulence-associated genes shown here are from the custom panel of APEC-associated genes. Members of the panel not represented here were not present in any isolates. Plasmid replicon and predicted antibiotic resistance gene data have been condensed to show major classes. Therefore, presence indicates ≥1 members of that class were detected. Full gene and plasmid detection data are available in Tables S5–S7.

Overall low virulence potential amongst isolates

To assess the potential for our isolates to cause disease in chickens we looked for genomes containing ≥4 of the virulence-associated genes iutA, hlyF, iss, iroN and ompT. This panel has been proposed to identify APEC more accurately than classical serotyping methods (Johnson et al., 2008). Within our isolates, Surrey013, Surrey017 and Surrey034 carried all five genes, while Surrey010 and Surrey050 carried all except iutA. However, genes are more strongly correlated with pathogenicity if they are carried on a plasmid rather than the chromosome (Johnson et al., 2008). Both Surrey013 and Surrey034 had plasmid-encoded copies of four of the genes, qualifying them as APEC. Surrey017 and Surrey050 carried four plasmid-encoded virulence-associated genes each, but not all were from the panel of five genes so are not counted as APEC under these criteria (Table S6). A further 16 strains carried one or two virulence-associated genes on plasmids.

An alternative diagnostic strategy defines APEC by categorization into any of four associations of virulence (Schouler et al., 2012). By these criteria 5/81 strains are categorized as APEC: Surrey034 (iutA+, P(F11)−, frzorf4+), Surrey001, Surrey002, Surrey014 and Surrey015 (iutA−, sitA+, aec26+). Surrey013 and Surrey017 might additionally be considered marginal as they were only discounted due to their predicted O8 serotype, rather than O78. Therefore, the two diagnostic measures combined indicate that 6/81 strains are APEC and 3/81 have increased virulence potential but fall short of the cut-off. Overall, only a minority of our isolates are likely to be APEC according to their complement of genetic virulence determinants.

APEC fall into the wider pathotype of Extraintestinal Pathogenic E. coli (ExPEC), which includes named groups causing disease in humans: Neonatal Meningitis E. coli (NMEC), Sepsis-Associated E. coli (SEPEC) and Uropathogenic E. coli (UPEC) (Sarowska et al., 2019). APEC may be closely related to human ExPEC and may act as a reservoir for ExPEC virulence-associated genes (Cummins et al., 2019). Therefore, to assess the wider pathogenicity potential of our strains, we compiled a panel of 24 virulence-associated genes. These included the APEC-associated genes above and other typical ExPEC virulence-associated genes (Table 4). Sixteen of these genes were present in at least one isolate (Fig. 5). Three isolates (Surrey031, Surrey045 and Surrey062) contained none of the genes and Surrey013 contained the most with 14/24 detected. The modal isolate contained two genes from the panel, and the median contained three genes.

Table 4 Members of the panel of virulence-associated genes used to predict virulence potential of isolates in this study.

Gene	Function	Protein	N° isolates	Ref.	
afa/dra	Adhesion	Afimbrial adhesin	0	(Johnson et al., 2003)	
fimH	Type 1 D-mannose specific adhesin	70	(Cummins et al., 2019)	
papA/felA	P fimbrial adhesin (type F11)	0	(Kariyawasam & Nolan, 2011; Schouler et al., 2012)	
papC	Outer membrane usher	0	(Dodson et al., 1993; Schouler et al., 2012)	
papG	P fimbrial tip adhesin	0	(Johnson & Brown, 1996; Schouler et al., 2012)	
sfa/foc	S fimbrial adhesin	0	(Johnson et al., 2003)	
iss	Host evasion	Increased serum survival lipoprotein	30	(Johnson et al., 2008)	
kpsM	Polysialic acid transport protein	15	(Stromberg et al., 2017)	
ompT	Outer membrane protease	32	(Johnson et al., 2008)	
traT	Complement resistance protein	19	(Cummins et al., 2019)	
chuA	Iron acquisition	TonB-dependent heme receptor	24	(Spurbeck et al., 2012)	
fyuA	Siderophore yersiniabactin receptor	9	(Stromberg et al., 2017)	
ireA	Iron-regulated outer membrane virulence protein	1	(Cummins et al., 2019)	
iroN	Siderophore salmochelin receptor	8	(Johnson et al., 2008)	
irp2	Yersiniabactin biosynthetic protein	8	(Cummins et al., 2019)	
iucD	L-lysine N6-monooxygenase (aerobactin synthesis)	3	(Cummins et al., 2019)	
iutA	Siderophore aerobactin receptor	3	(Johnson et al., 2008; Schouler et al., 2012)	
sitA	Fe/Mn ABC transporter substrate binding	26	(Schouler et al., 2012)	
icmH/aec26	Secretion	type IVB secretion system protein	23	(De Pace et al., 2010; Schouler et al., 2012)	
frz orf4	Sugar transport	PTS fructose transporter subunit IIC	1	(Rouquet et al., 2009; Schouler et al., 2012)	
astA	Toxin	Heat-stable enterotoxin EAST1	15	(Kaper, Nataro & Mobley, 2004)	
hlyF	SDR Family oxidoreductase	8	(Johnson et al., 2008)	
tsh	Temperature-sensitive haemagglutinin	0	(Cummins et al., 2019)	
vat	Vacuolated autotransporter toxin	0	(Spurbeck et al., 2012)	

Other commonly identified virulence-associated genes from comparison with the ecoli_VF database (a much larger database covering E. coli in general) included fimA–I, yagV–Z, ompA, entA–F/S, fepA–D/G and fes. Genes associated with the general secretory pathway (gspC–M) (Francetic & Pugsley, 1996) and type III secretion systems 1 and 2 (espL, R, X, Y; eivA, C, E–G, I, J) were also frequently identified (Franzin & Sircili, 2015; Fox et al., 2020). Many of these genes are associated with intestinal E. coli infections. Full predicted virulence-associated gene content of all strains, from comparison with the ecoli_VF database are available in Table S7.

Low carriage of predicted antimicrobial resistance genes

From our comparison with the NCBI AMRFinderPlus database, we predicted low levels of antimicrobial resistance. Among the isolates, 62/81 had no predicted resistance genes (Table S8). However, five isolates carried three predicted resistance genes, one carried two genes, and 13 carried one gene. Fourteen isolates carried a gene for predicted tetracycline resistance, nine carried a predicted aminoglycoside resistance gene, three carried a predicted β-lactamase class bla-TEM gene, two carried a predicted trimethoprim resistance gene, and predicted genes for fosfomycin and sulfonamide resistance were found in a single isolate each (Fig. 5). Every isolate except three carried a single copy of a β-lactamase class blaEC gene (one of blaEC, blaEC8, blaEC13, blaEC15, blaEC18 or blaEC19). These genes are frequently detected in genomes of β-lactam-susceptible isolates and therefore were discounted from the analysis. We did not experimentally verify the predicted resistance phenotypes; however, a preliminary screening following colony isolation also revealed low levels of phenotypic resistance to nine antibiotics (Table S1).

Discussion

Our data reveal remarkable diversity within the genus Escherichia from such a commonplace setting and small sample size. After minimal selection during sample collection, we were able to recapitulate almost the entire phylogeny of Escherichia. However, our initial selection for E. coli identity may still have excluded some isolates with phenotypes that differ from typical E. coli. On the other hand, the isolation of so many other Escherichia species raises doubts over the specificity of CHROMagar Orientation Medium for Escherichia coli sensu stricto. This finding may have implications for the use of CHROMagar Orientation Medium in clinical research and diagnostics, and warrants further investigation into the growth and morphology of different members of the Escherichia genus on this medium. We confirmed the diversity and identity of the isolates by several different methods, including PCR, in silico phylotyping and MLST, and core genome analysis. Our final core genome-derived phylogenetic tree was also in agreement with previously published phylogenies for the genus (Zhou et al., 2020; Abram et al., 2021; Gilroy et al., 2021).

The generally low carriage of virulence-associated genes within the isolates suggests that the majority are non-pathogenic and more likely represent commensal members of the gut microbiota (Wigley, 2015). By extension, this also supports the idea that although many strains are capable of opportunistic infection, most frank pathogens in the Escherichia genus belong to relatively well-defined pathotypes (Johnson et al., 2008). In support of this conclusion, three key phylogroups, B2, G and clade I, which have been associated with virulence in poultry (Papouskova et al., 2020; Mehat, van Vliet & La Ragione, 2021), were missing from our collection. This could be due to the relatively small sample size and small flock or might reflect that we only sampled healthy birds. Equally, the most common phylogroup in this study was phylogroup A, which has been linked to commensalism in chickens and other omnivores (Smati et al., 2015). E. albertii and Shigella were also absent from our collection. They are less well studied in birds, but have also both been linked to virulence in poultry (Shi et al., 2014; Gomes et al., 2020). We cannot rule out the possibility that APEC strains were simply lower in abundance and so were overlooked during culturing. As we only sampled healthy birds, it seems likely that this is a true reflection of the prevalence of these phylogroups among our study population. The limitations of our study prevent us from inferring any general conclusions about the distribution of Escherichia strains within chickens in other settings. Nevertheless, if such striking diversity is present in a few birds in a single location, it surely suggests that similar diversity (although perhaps with different population structures) has so far been overlooked elsewhere.

Although we did not track individual birds temporally and so cannot verify stable colonization of isolates within birds, most fecal samples yielded multiple distinct isolates. There were no clear indications of niche adaptation based on the pangenome and plasmid content of the strains, as gene clusters and plasmid replicons were spread fairly evenly among phylogenomic groups, and most phylogroups contained few isolates. The use of only short-read data also limited the completeness of our assemblies and our ability to identify plasmids and plasmid-encoded genes (Arredondo-Alonso et al., 2017). Further work involving more isolates and using both short- and long-read data will be necessary to identify unique, characteristic genetic components for each species and phylogroup, and to establish their spatial and functional niches within the chicken gut.

Recent studies have found similarly diverse populations of Escherichia after sampling of healthy pigs (Ahmed, Olsen & Herrero-Fresno, 2017; Bok et al., 2020), cattle and sheep (Shaw et al., 2021), and non-human primates (Foster-Nyarko et al., 2020), although it is notable that representatives of the ‘cryptic clades’ are scarce within these data. The reasons for this scarcity are unclear but are likely to include the use of different selection methods and the focus on virulent/AMR strains and E. coli sensu stricto. Escherichia diversity varies among samples from healthy humans (Bok et al., 2018; Foster-Nyarko et al., 2021a). High Escherichia species diversity has been detected within chickens previously (Vounba et al., 2019a; Foster-Nyarko et al., 2021b), although not from such a narrow pool of host birds as in the current study. Members of the cryptic clades have also previously been found in relatively high abundance in wild birds of various species (Clermont et al., 2011) but a definitive link to chickens has not been established. A recent metagenomics study included 60 different fecal samples from the same birds used here. Culturing of 3 of those samples yielded isolates of E. marmotae, E. whittamii and phylogroups A, B1, C, D and E, reflecting the diversity found in this study (Gilroy et al., 2021). However, most studies in chickens have taken samples from diseased birds and/or have focused on antimicrobial resistance and pathogenicity of the isolates recovered (Braga et al., 2016; Cummins et al., 2019; Vounba et al., 2019b; Papouskova et al., 2020; Kubelová et al., 2021).

Conclusions

The dearth of whole-genome sequencing studies that include all members of Escherichia (and sufficient metadata) skews our understanding of the genus and makes it impossible to discern patterns within or between sampling environments, geographical regions, diets, health status and so on. Consequently, there is much still to learn about the population structure and ecology of Escherichia (Lagerstrom & Hadly, 2021). We were surprised to find that isolates representing almost the entire known phylogeny of Escherichia were recovered from fecal samples from a small flock of healthy layer chickens, including species that have only recently been recognized (previously known as cryptic clades) and for which chicken-associated isolates have not been widely reported. These isolates had low carriage rates of antimicrobial resistance genes and virulence factors, suggesting that similar isolates might often be overlooked in studies that focus on these traits. Our findings highlight the surprising diversity of Escherichia harbored by even an individual chicken and emphasize the need to broaden the focus of research to encapsulate the full variety of species.

Supplemental Information

Supplemental Information 1 Neighbor-joining phylogenomic tree for 81 Escherichia isolates from healthy chicken fecal samples.

The same tree is shown with nodes colored by (A) predicted phylogroup according to the program ClermonTyping, and (B) month in which the fecal sample was collected. Isolates cluster strongly with their predicted phylogroups, and isolates from all groups were recovered throughout the experiment.

Click here for additional data file.

Supplemental Information 2 Gene sequences used to construct a custom ABRicate database for detection of 24 APEC-related virulence genes.

Genes were identified based on a literature search and sequences were obtained from the ecoli_VF and NCBI Nucleotide databases. The rpoS gene from E. coli MG1655 was included as a positive control for detection.

Click here for additional data file.

Supplemental Information 3 Metadata associated with 100 Escherichia isolates from faecal samples of healthy Lohmann Brown layer hens.

Isolate name: strain name used for this paper.

UoS isolate ID: alternative strain ID used for internal cataloguing.

2ndry ID: alternative ID describing multiple strains isolated from individual chickens (Chicken#-isolate#).

ND: not detected.

*: multiple alleles detected, possibly from split contig within gene.

Click here for additional data file.

Supplemental Information 4 Source information and accession numbers for 33 reference strains from avian hosts, used in the construction of a core genome alignment for the Escherichia genus.

Click here for additional data file.

Supplemental Information 5 HierCC HC0 level clusters of the original 100 isolates with genome quality scores.

The isolate with the highest genome quality score in each cluster was used as the representative for that cluster.

Click here for additional data file.

Supplemental Information 6 Functional predictions for all pangenome gene clusters.

Click here for additional data file.

Supplemental Information 7 Summary output from ABRicate using the PlasmidFinder database for 81 isolates of Escherichia..

Parameters used were: minimum coverage = 90% and minimum identity = 80%. Isolates that do not appear had no matching genes using these search parameters. Gene presence is denoted by its coverage (%). Gene absence is represented by a full stop.

Click here for additional data file.

Supplemental Information 8 Summary output from ABRicate using our custom virulence-associated gene database for plasmid-related contigs from 81 isolates of Escherichia.

Contigs were identified as chromosome- or plasmid-associated using platon v.1.6, and all plasmid-associated contigs wer eused as input for ABRicate. Parameters used were: minimum coverage = 90% and minimum identity = 80%. Gene presence is denoted by its coverage (%). Gene absence is represented by a full stop. Only samples and genes with hits are included in the table.

Click here for additional data file.

Supplemental Information 9 Summary output from ABRicate using the ecoli_VF database for 81 isolates of Escherichia.

Parameters used were: minimum coverage = 90% and minimum identity = 80%. Gene presence is denoted by its coverage (%). Gene absence is represented by a full stop.

Click here for additional data file.

Supplemental Information 10 Summary output from ABRicate using the NCBI AMRFinderPlus database for 81 isolates of Escherichia.

Parameters used were: minimum coverage = 90% and minimum identity = 80%. Isolates that do not appear had no matching genes using these search parameters. Gene presence is denoted by its coverage (%). Gene absence is represented by a full stop.

Click here for additional data file.

We thank Mr Steven Rudder and Mr David Baker for their assistance with sequencing, and the farm for assisting with the study.

Additional Information and Declarations

Competing Interests

Author Contributions

Animal Ethics

DNA Deposition

Data Availability

The authors declare that there are no competing interests.

Nicholas M. Thomson conceived and designed the experiments, performed the experiments, analyzed the data, prepared figures and/or tables, authored or reviewed drafts of the paper, and approved the final draft.

Rachel Gilroy conceived and designed the experiments, performed the experiments, analyzed the data, prepared figures and/or tables, authored or reviewed drafts of the paper, and approved the final draft.

Maria Getino conceived and designed the experiments, performed the experiments, analyzed the data, prepared figures and/or tables, authored or reviewed drafts of the paper, and approved the final draft.

Ebenezer Foster-Nyarko analyzed the data, prepared figures and/or tables, authored or reviewed drafts of the paper, and approved the final draft.

Arnoud H.M. van Vliet analyzed the data, authored or reviewed drafts of the paper, and approved the final draft.

Roberto M. La Ragione conceived and designed the experiments, authored or reviewed drafts of the paper, and approved the final draft.

Mark J. Pallen conceived and designed the experiments, authored or reviewed drafts of the paper, and approved the final draft.

The following information was supplied relating to ethical approvals (i.e., approving body and any reference numbers):

The study was approved by the University of Surrey’s NASPA Ethical Review Assessment Committee with project number NERA-2018-011.

The following information was supplied regarding the deposition of DNA sequences:

The raw sequencing reads and assemblies used in this work are available at NCBI SRA: PRJNA757375.

The individual accession numbers of the sequencing reads and assemblies for the isolates are available in Table S1.

The following information was supplied regarding data availability:

The raw data is available in the Supplemental Files.

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
