# Peer review of "Remarkable genomic diversity among Escherichia isolates recovered from healthy chickens"

_PeerJ, doi:10.7717/peerj.12935_

## Round 0.1 · original submission · Major Revisions

Both reviewers were overall positive about the manuscript, and its findings, but felt that there were improvements that could be made. These can be found below.

Reviewer 1 ·

Basic reporting

The goal of this manuscript was to survey the diversity of Escherichia in a single layer flock using a combination of traditional isolation coupled with genomic approaches. The authors found that, by screening a population of ~100 isolates, there was a very diverse population of Escherichia harbored by a relatively small number of birds. These isolates appeared to be primarily non-pathogenic, as they lacked virulence genes typical of ExPEC and APEC, and they no isolates belonged to the notorious B2 phylogenetic type typical of ExPEC. Overall, this work is a nice snapshot of the diversity of the genus harbored by birds and is useful information.

Experimental design

Can the authors please describe the farm used to obtain the Lohmann layers? Some more information would be relevant because the focus is on Escherichia diversity and ecology, such as flock size, barn design, ventilation, and were the pullets for this flock sourced from multiple flocks or a single flock?

Some clarification on Table 1 would be helpful. Why did the number of primary isolates collected differ by date collected? Age of bird in the table would be helpful as well. Is number of unique isolates based on the GTG5 PCR? It was unclear if additional isolates were screened and filtered into those characterized in Table 1, or if this table represents all isolates screened?

For the custom APEC database, some of the genes listed are really not APEC virulence factors and are instead ExPEC virulence factors derived from the work of James Johnson. I would suggest that this list of genes be refined according to more recent APEC literature, which focuses on some of the genes on the list but emphasizes the presence of the pathogenicity-associated island on ColV-like plasmids, which is established the strongest link between virulence potential in birds, and is of highest prevalence in clinical isolates and most differentiating between clinical versus commensal isolates.

Figure 1 could be used to further clarify where the GTG5 screen falls into the workflow?

Figure 2, definition of the colors should be clearly defined in the legend (what are they, specifically?). Also, no scale is provided for the tree.

Figure 4, title should be plasmid replicon contents, not plasmid contents as the approach used only surveyed replicons.

One of the most enlightening aspects of this manuscript was the E. coli ChromeAgar clearly grows other Escherichia. This should be further discussed in the manuscript or brought to attention of the reader as it has substantial implications for researchers.

While the primary focus of the manuscript was on the substantial diversity of E. coli even in a setting that is fairly uniform and controlled, the authors do discuss the absence of the B2 phylotype and its implications re: commensal E. coli populations in the layer gut. Since this was brought up, some limitations of the study need to be addressed in the study, including N=1 flock, small numbers of isolates and timepoints, small number of birds, etc.

Is there any way to better present the diversity of Escherichia harbored by a single bird at a single timepoint? This would be useful.

Validity of the findings

The findings of the authors that extensive diversity exists were supported by the genomic data. I question if the study design was robust enough to make any conclusions about the ecological makeup of Escherichia in the layer gut, that could be generalized more broadly. I would suggest that the authors take caution in some of the discussion points focusing on APEC in the gut, and how an N=1 flock translates to layers wordwide.

·

Basic reporting

The paper of Thomson et al. analyses the genome sequences of 100 isolates of Escherichia coli from fecal samples of 24 healthy adult chicken. The work is globally well done. The main finding is that, in this small sample size of hosts, a huge genomic diversity is found with representative of the major phylogenetic lineages of the genus Escherichia except E. albertii, Shigella and E. coli phylogroups B2 and G. This is indeed remarkable, as stated in the title.

Here are some remarks to improve the manuscript.

The authors should look to the serotype (O:H) and fimH diversity of the strains. They should also search for intestinal virulence genes.

The authors should discuss the fact that APEC strains are mostly found in G (ST117) and B2 (ST95) phylogroups, which are absent from their data set.

Lines 284-85, the authors are right, but there are some tools that can help and should be tested (see doi: 10.1099/mgen.0.000398).

Lines 366-67, not true. See Clermont et al. Environ Microbiol, 2011, clades are frequently isolated in birds.

Lines 28-29, please rephrase in a more comprehensive way without abbreviation.

Line 114, LB is for Lysogeny Broth (not Luria Bertani).

Experimental design

OK

Validity of the findings

OK

---

## Round 0.2 · accepted · Accept

Both reviewers were happy that you had fully addressed all concerns that they had, so this can now be accepted for publication.

Reviewer 1 ·

Basic reporting

The authors have addressed all concerns of reviewers #1 and #2 nicely. I have no further concerns. This is a nice and useful paper.

Experimental design

The authors have addressed all concerns of reviewers #1 and #2 nicely.

Validity of the findings

The authors have addressed all concerns of reviewers #1 and #2 nicely.

·

Basic reporting

The paper has been improved and can be published.

Experimental design

OK

Validity of the findings

OK